# OpenReview forum: "Aligning Large Language Models with Representation Editing: A Control Perspective"
_NeurIPS.cc/2024/Conference — NeurIPS 2024 poster_

### Official Review · Reviewer_F1xo · 2024-06-12

**Soundness:** 4
**Presentation:** 2
**Contribution:** 3
**Rating:** 7
**Confidence:** 3

**Summary:**

In their paper, the authors introduce RE-CONTROL, a novel approach designed to align Large Language Models (LLMs) through representation editing. They view LLMs as discrete-time stochastic dynamical systems and propose the insertion of control signals into the internal representations. This technique allows for precise manipulation of the model's outputs during test time token by token. The experiments show that this method increases the win rate on HH dataset and does not need significant inference time.

**Strengths:**

- Viewing LLMs as a dynamical system and interpret the steering vector as a kind of controlling signal to align models is innovative.
- Make LLMs adjustable during the generation process, and the evaluation does not have to wait until the entire sentence is generated.
- They empirically show that their method outperform some test-time alignment methods and does not need significant inference time, which makes the method be more practical usable.

**Weaknesses:**

- Some parts of the paper are confusing, especially certain expressions. For example, they did not clarify some notations like a_t, V_{phi} etc.. The legend in figure 1 seems mismatched. And some figures are not mentioned in the paper.
- I think the performance of this method is highly depend on the value model. However, the paper does not discuss the reliability of the value model, which is crucial since it needs to assess the alignment effectiveness of the entire result based on each newly generated token and do so before the results are generated.
- The theoretical analysis and interpretation of their method is interesting, but lack rigor. e.g. the generated token (y_t) should be determined by logits (o_t), which is a part of state in the dynamic system. However, the paper interprets the generated token as kind of random variable or random noise (w_t).

**Questions:**

Please refer to the part of weaknesses.

**Limitations:**

Limitation are sufficiently discussed.

---

> ### Author Rebuttal · Authors · 2024-08-07
>
> **Q1: Some parts of the paper are confusing, especially certain expressions. For example, they did not clarify some notations like a_t, V_{phi} etc.. The legend in figure 1 seems mismatched. And some figures are not mentioned in the paper.**
>
> A: $a_t$ is a typo; we meant $u_t$, which is the control signal. $V_{\phi}$ represents the value function parameterized by a neural network with parameters \(\phi\). We will correct this typo and add more illustrations of the notations in our revised version. Additionally, we will polish the figures.
>
> **Q2: I think the performance of this method is highly depend on the value model. However, the paper does not discuss the reliability of the value model, which is crucial since it needs to assess the alignment effectiveness of the entire result based on each newly generated token and do so before the results are generated.**
>
> A: There are three pieces of evidence that demonstrate the reliability of our value model. First, comprehensive evaluation metrics show that RE-Control outperforms the baselines. Second, we test RE-Control and the most competitive baselines on out-of-distribution data. Figure 4 illustrates that our method generalizes well to new input distributions beyond the training distribution of the value function. Third, the uploaded PDF in the general response shows the validation loss during the training process, which is very smooth. This further indicates the reliability of our value model training.
>
> **Q3: The theoretical analysis and interpretation of their method is interesting, but lack rigor. e.g. the generated token (y_t) should be determined by logits (o_t), which is a part of state in the dynamic system. However, the paper interprets the generated token as kind of random variable or random noise (w_t).**
>
> A: This depends on how one defines the language dynamical system, and we believe both perspectives are mathematically correct. In our work, we emphasize representation editing, adding control signals only to the representation space, thus defining the system's state within this space. Mathematically, the generated token functions similarly to the random component in traditional stochastic dynamical systems, as both are functions of the state. Indeed, one could also define the generated token as part of the state. We will include this alternative view in our revised version.

---

> > ### Comment · Reviewer_F1xo · 2024-08-13
> >
> > Thank you for your responses. It addressed some questions. I will maintain my score.

---

### Official Review · Reviewer_vtBo · 2024-07-11

**Soundness:** 3
**Presentation:** 3
**Contribution:** 2
**Rating:** 6
**Confidence:** 4

**Summary:**

The paper suggests editing language model features for alignment tasks. The authors first learn a value function of a language model from a human-preference dataset. They then increment feature representations in model layers to maximize test-time utility. Empirical evidence shows that this feature editing method surpasses both test-time and training-time alignment baselines.

**Strengths:**

The proposed method, RE-CONTROL, is a useful middle ground between current training-time and test-time alignment methods:

- RE-CONTROL, unlike existing training-time methods, does not alter a language model’s parameters, reducing training costs. Instead, it learns a value function offline.

- RE-CONTROL, unlike existing test-time methods, employs a learned value function to inject feature increments into features of language models.

The experiments are extensive in that they compared RE-CONTROL with both training-time and test-time alignment methods.

**Weaknesses:**

While the paper is technically well-executed, I believe it has three main limitations:  (i) the lack of compute--performance tradeoff analysis (ii) the lack of details in comparing RE-CONTROL with training-time alignment baselines. (iii) the limitation in application scope.

First, a compute-performance tradeoff analysis would clarify the behavior of RE-CONTROL. RE-CONTROL is more compute-intensive than other test-time decoding alternatives because it requires gradient ascent steps at decoding time (Section 4.4). These steps add up and can become quite intensive for generating long text. Therefore, comparing RE-CONTROL with test-time alignment alternatives while considering compute time would be informative. For instance, the authors could display the win rate of different test-time decoding methods on the y-axis and their wallclock time on the x-axis.

Second, I think the performance comparison between RE-CONTROL and training-time alignment methods in Section 6.1 seems very preliminary. There, the authors empirically show that the test-time alignment method RE-CONTROL *outperforms* training-time alignment methods like PPO, by concluding that

>We observe that RE-CONTROL achieves a higher GPT-4 win rate and average reward compared to both PPO and DPO. Furthermore, RE-CONTROL also outperforms these methods in terms of diversity and coherence.

I'm puzzled by how to interpret the results here. Should the take-home message here be "Decoding-time RE-CONTROL is better than training-time PPO in alignment. Period." or are there qualifications to this statement? I strongly suspect that some qualification is needed. To some extent, RE-CONTROL is a decoding-time approximation of PPO. Both methods use a learned value function to steer the model's behavior. At decoding time, RE-CONTROL does this in a more lossy (due to test-time gradient ascent) and shallower (because not all parameters are updated) way. Thus, with adequate training, I expected PPO to yield better results than RE-CONTROL. Note that this doesn't undermine RE-CONTROL's capability, as it is more lightweight than PPO.


Thirdly, while RE-CONTROL is technically sound, its application scope seems narrow. To my understanding, RE-CONTROL is most appealing to users who are unwilling to train a language model offline, who are willing to train a value function offline, who aim to save computing power during training, and who don't mind using more compute during decoding. These intersections of users seem limiting. This raises the question: Is it better to simply use a similar compute budget for efficient alignment (e.g., LoRa) of the LM model using standard methods (DPO, PPO, etc.) and avoid ongoing compute costs during decoding?

**Questions:**

As mentioned above, in my opinion, it is surprising that decoding-time RE-CONTROL outperforms training-time PPO. To compare PPO and RE-CONTROL more carefully, could the authors consider some ablation studies? For example, you could use the same value function for both PPO and RE-CONTROL, one at training time to fine-tune the model parameters and the other at decoding time to produce the feature increment and compare the results.

**Limitations:**

See the "Weaknesses" section above.

---

> ### Author Rebuttal · Authors · 2024-08-07
>
> **Q1. First, a compute-performance tradeoff analysis would clarify the behavior of RE-CONTROL. RE-CONTROL is more compute-intensive than other test-time decoding alternatives because it requires gradient ascent steps at decoding time (Section 4.4). These steps add up and can become quite intensive for generating long text. Therefore, comparing RE-CONTROL with test-time alignment alternatives while considering compute time would be informative. For instance, the authors could display the win rate of different test-time decoding methods on the y-axis and their wallclock time on the x-axis.**
>
> A: Actually, RE-Control is significantly faster than controlled decoding. We compared the inference times of all methods using the best hyperparameters selected on the validation set in our paper. Please refer to the inference time (in hours) column in Table 1 of our paper. As shown, ARGS ([1], referred to as controlled decoding in the paper) and controlled decoding with a value function [2] are much slower than RE-Control. This discrepancy arises because controlled decoding needs to evaluate multiple candidate tokens and repeatedly perform forward passes through the entire reward model. In contrast, RE-Control only requires optimization through a value function, which is a two- or three-layer neural network, making it significantly faster.
>
> Additionally, we provide more analysis on the inference speed in the general response. Specifically, we plot the inference time under different batch sizes and the compute-performance tradeoff. The results further verify that RE-Control is significantly faster than ARGS. Please see the uploaded PDF and the general response for a detailed discussion.
>
> **Q2. I'm puzzled by how to interpret the results here. Should the take-home message here be "Decoding-time RE-CONTROL is better than training-time PPO in alignment. Period." or are there qualifications to this statement? ... At decoding time, RE-CONTROL does this in a more lossy (due to test-time gradient ascent) and shallower (because not all parameters are updated) way. Thus, with adequate training, I expected PPO to yield better results than RE-CONTROL. Note that this doesn't undermine RE-CONTROL's capability, as it is more lightweight than PPO.**
>
> A: First, since our paper focuses on test-time alignment, our comparison is primarily with other test-time alignment methods. When comparing with PPO and DPO, we follow the setup in ARGS [1]. Table 4 in Section 4 of [1] demonstrates that ARGS can outperform both PPO and DPO. Since RE-Control outperforms ARGS, it is unsurprising that RE-Control slightly outperforms PPO and DPO as well. We suspect this is because we use LoRA for PPO and DPO, similar to the ARGS paper, which introduces approximation errors. However, we lack the computational resources to perform a direct comparison with full fine-tuning and this is exactly the motivation why we want test-time alignment methods.
>
> Second, we want to clarify that RE-Control does not update model parameters at test time but optimizes the hidden representation directly. Existing works on representation editing [3,4] also show that it can outperform fine-tuning methods for other tasks.
>
> That said, we agree that it is not appropriate to conclude that test-time alignment is better. Therefore, in lines 296-297, we state that RE-Control can achieve competitive performance compared to PPO and DPO. We will further soften our arguments in the revised version, emphasizing that the use of LoRA for PPO and DPO may introduce additional approximation errors. Additionally, we will add a discussion on when to use test-time alignment methods versus fine-tuning methods, as addressed in Q3 below.
>
> **Q3. Thirdly, while RE-CONTROL is technically sound, its application scope seems narrow. ...This raises the question: Is it better to simply use a similar compute budget for efficient alignment (e.g., LoRa) of the LM model using standard methods (DPO, PPO, etc.) and avoid ongoing compute costs during decoding?**
>
> A: This is a broad question: do we need test-time alignment methods? Given the "no free lunch" principle, test-time alignment methods like guided decoding indeed increase inference time. However, as we mentioned in Q1, RE-Control has already shown a significant speed-up in inference time compared to controlled decoding. This represents an important advancement that can make test-alignment practical in real world applications.
>
> We also want to highlight that even with LoRA, fine-tuning methods still require substantially more computing resources. For example, in our experiments, we used LoRA with rank 8 for both PPO and DPO. PPO training takes about 3 days on three A100 GPUs, while DPO takes about 1.5 days on three A100 GPUs. In contrast, training the value function only takes around 3 hours on one A100 GPU.
>
> For users who prioritize real-time inference, amortizing the computation into the training process might still be preferable. However, for those without the resources to fine-tune at all, test-time alignment is a better choice and can easily adapt to different alignment objectives. We will include this discussion in our revised version.
>
> **Q4. To compare PPO and RE-CONTROL more carefully, ... one at training time to fine-tune the model parameters and the other at decoding time to produce the feature increment and compare the results.**
>
> In PPO, it also needs to train a value function to estimate the advantage function. However, since PPO is an online algorithm that requires iterative training of the value function, it is hard to employ a pre-trained value function.
>
> ## References:
>
> [1] ARGS: Alignment As Reward-Guided Search, ICLR 2024
>
> [2] Controlled Decoding from Language Models, ICML 2024
>
> [3] Inference-Time Intervention: Eliciting Truthful Answers from a Language Model, NeurIPS 2023
>
> [4] ReFT: Representation Finetuning for Language Models, Arixv

---

> > ### Comment · Reviewer_vtBo · 2024-08-12
> > **Response**
> >
> > Thank you for your response. It addressed some of my questions, and I raised my score.
> >
> > > A: Actually, RE-Control is significantly faster than controlled decoding. We compared the inference times of all methods using the best hyperparameters selected on the validation set in our paper. Please refer to the inference time (in hours) column in Table 1 of our paper.
> >
> > Apologies for overlooking the inference time mentioned in Table 1. The results are impressive!
> >
> > > We suspect this is because we use LoRA for PPO and DPO, similar to the ARGS paper, which introduces approximation errors. However, we lack the computational resources to perform a direct comparison with full fine-tuning and this is exactly the motivation why we want test-time alignment methods.
> >
> > Thank you for the clarification. As the authors mentioned, I recommend they add a qualification to their statements in Section 6.1, such as: "Overall, the results indicate that our approach is a competitive alternative to **parameter-efficient / LoRa-based** training-time alignment methods." Without this qualification, readers might question the rigor of the experimental results.
> >
> > > We also want to highlight that even with LoRA, fine-tuning methods still require substantially more computing resources. For example, in our experiments, we used LoRA with rank 8 for both PPO and DPO. PPO training takes about 3 days on three A100 GPUs, while DPO takes about 1.5 days on three A100 GPUs. In contrast, training the value function only takes around 3 hours on one A100 GPU.
> >
> > I see. These numbers are very impressive. I suggest the authors include a similar discussion in their paper or code repository, as doing so will make the method more appealing to practitioners.
> >
> > > However, since PPO is an online algorithm that requires iterative training of the value function, it is hard to employ a pre-trained value function.
> >
> > I'm not sure I understand this. Wouldn't it be possible to take a value function trained with PPO and use it within the authors' decoding framework? This approach could allow for a more direct comparison between PPO and the authors' method, as they use the same underlying value function. A similar approach for reusing the PPO value function is discussed here: https://openreview.net/forum?id=QaODpeRaOK.

---

> ### Author Response · Authors · 2024-08-13
> **Thanks for your reply!**
>
> Thank you for your reply and for raising the score! We would like to provide some comments as follows:
>
> **Thank you for the clarification. As the authors mentioned, I recommend they add a qualification to their statements in Section 6.1, such as: "Overall, the results indicate that our approach is a competitive alternative to parameter-efficient / LoRa-based training-time alignment methods." Without this qualification, readers might question the rigor of the experimental results.**
>
> Thank you for the suggestion. We will explicitly clarify that our comparisons are focused on LoRa-based training-time alignment methods, and we demonstrate competitive performance relative to them.
>
> **I suggest the authors include a similar discussion in their paper or code repository, as doing so will make the method more appealing to practitioners.**
>
> Thank you for your suggestion. We will include the training time in our comparisons with LoRa-based training-time alignment methods in the revised version.
>
> **I'm not sure I understand this. Wouldn't it be possible to take a value function trained with PPO and use it within the authors' decoding framework? This approach could allow for a more direct comparison between PPO and the authors' method, as they use the same underlying value function. A similar approach for reusing the PPO value function is discussed here: https://openreview.net/forum?id=QaODpeRaOK.**
>
> Thank you for suggesting this paper. The training objective of the value function in PPO is slightly different from ours. In PPO (as  Equation 1 in your suggested paper), regularization is incorporated into the reward function. In contrast, our value function estimates the true reward (see Line 168), and we introduce regularization during testing time by tuning the hyperparameters, such as the number of iterations and step size. This difference means that using the PPO value function could complicate hyperparameter tuning, as it would require retraining the value function each time we adjust the regularization strength.
>
> That said, we agree that it would be interesting to explore the performance of using the value function from PPO. Additionally, examining how the PPO value function performs in baseline-controlled decoding scenarios would also be interesting.
>
> However, with only about a day remaining in the discussion period, it would be challenging to conduct these new experiments, especially since tuning the hyperparameters in this context would require retraining the PPO model. We plan to explore this further in our camera-ready version and hope this is understandable to you. Moreover, since our primary focus is on test-time alignment methods, we believe this does not affect our main conclusions.
>
> Again, Thanks for this interesting suggestion!
>
> Best,
>
> RE-Control authors

---

### Official Review · Reviewer_XYhX · 2024-07-12

**Soundness:** 2
**Presentation:** 3
**Contribution:** 2
**Rating:** 5
**Confidence:** 3

**Summary:**

The paper introduces an alternative procedure for LLM alignment that does not fine-tune LLM weights, but instead learns a separate value function that is used to update hidden states. The value function is learned using a variation of temporal difference, then applied at inference time to modify hidden states by gradient ascent, maximizing the predicted state value. Authors evaluate their approach with multiple 7B LLMs on HH-RLHF data, comparing against both RLHF and training-free baselines. The paper also analyzes OOD generalization to HarmfulQA.

**Strengths:**

- Authors propose an interesting approach to that can be used to alter LLM behavior in general
- When experimenting with HH-RLHF dataset, authors evaluate against multiple types of baselines and provide additional analysis that was interesting to read
- The paper is generally well-written and easy to follow
- Authors made the code available, in a (mostly) serviceable state

**Weaknesses:**

**1a. Motivation for the choice of baselines.**

In your work, you cite, among others, ARGS[26], DeAL [22], Value Augmented Sampling [21] that also learn value functions and use them to steer model outputs (in other ways), but, to the best of my knowledge, you do not compare against them as baselines, instead choosing a relatively older work on controlled decoding. While [21] may be dismissed as concurrent work, the other works appear to be a relevant alternative and it is not clear why they were not chosen as baselines.

If there is a reason why these works will, beyond reasonable doubt, fail at the task that you evaluate on, I would recommend that you explain this in the paper. If there is no such reason, the paper would benefit from comparing against them.

**1b. Motivation for the choice of models**

Your paper focuses on Llama, Vicuna and Falcon models, of the 7B variety. While these are indeed LLMs, the original Llama was released circa 1.5 years ago and since then, LLMs improved **significantly** across tasks.
Picking older LLMs appears counterintuitive, as their generally worse quality makes it harder to measure possible drawdowns introduced by LLM alignment.

If you have a reason for choosing these models, please explain why you focus on older LLMs as compared to, for example, Llama 3 8B (or 70B), Qwen2, Gemma or other models near the top of https://huggingface.co/spaces/open-llm-leaderboard/open_llm_leaderboard . If there is no such reason, the paper would benefit from switching to more accurate models.

**2. Inference time exploration**

LLM use cases are often sensitive to inference throughput (tokens per second) and latency (time to first / next token).

To the best of my understanding, RE-Control introduces an iterative optimization step to each forward pass during autoregressive inference. Depending on the configuration, this may result in a significant slowdown, which may limit the practical significance of your approach.

I would argue that the work would benefit from analyzing this difference in speed in different settings (e.g. single-sequence vs batch inference, etc).

**3.  Main experiments are limited to one dataset and relatively small past generation LLMs, ranked by GPT-4**

This is definitely not a fault on authors' side, but the paper makes its main conclusions based on 7B models, using reward functions trained on a single dataset. This could result in accidental false conclusions if it turns out that, for instance, RE-Control harms the quality of stronger models or if it is somehow implicitly overfitting on on GPT4 opinions.

The standard way to minimize this risk is to diversify the experiments: try alternative alignment datasets (e.g. webgpt_comparisons, oasst1, etc), try larger models (llama-3 70B), introduce human rankings in some setups, etc. I understand that not all of these evaluations may be available to the authors, but for a NeurIPS publication, I would expect more variation in the experiments and, if there is a confounder that could not be eliminated (e.g. using GPT4 and no human eval), it should be stated among the limitations section.

**Questions:**

**Questions on the definition of state**

To the best of my (possibly wrong) understanding, when you apply Bellman equation, you assume that the dynamic system's state satisfies Markov assumption. [If not, please explain why not]

Since LLMs use attention to previous hidden states, hidden vector for a specific state do not satisfy Markov assumption, since LLM's next token probability depends not only on them, but on a more distant past as well. In contrast, a fully markovian state would need to contain all previous hidden vectors, or the current hidden vectors and all past KV projections, or a sequence of all previous tokens  (no hidden vectors necessary).

In other words, **when you define V(s), does s refer to just the current token's hiddens or a full state with Markov assumption?**

If you mean the latter state, then the test-time intervention (S4.4) needs to modify all previous hidden states of an LLM. This is important because modifying past hidden states may result in a very inefficient LLM inference algorithm.

If only the current state, you seem to apply policy iteration (S4.2-4.3) to a non-markov state. Please explain how you make sure that this algorithm still has the guarantees of optimal policy. If it doesn't, please clearly explain that the algorithm is a heuristic inspired by PI rather than actual PI.

### On reproducibility

To reiterate, the fact that you publish the code is great. None of my complaints below affected the final score.

The codebase lacks library versions (requirements.txt / dockerfile / list them in the readme), which makes it difficult to reproduce, especially in the future. While I ultimately managed to run the code by choosing the libraries with an educated guess (and minor modifications to the code), I am still not sure if I got the method to work "as intended" and not introduce silent errors.

For legal reasons, it would be best to direct the users to a version of Llama 7B that contains its original license, at least in the final version of the paper.

Using GPT-4 opinion means that the experiments would be difficult to reproduce after it is cycled ou




### Typos / minor:

> L16 LLama

The capitalization for the first version was LLaMA, second and third are Llama.

> supplementary code: intervented_model

you may have meant “intervened”

**Limitations:**

The "Limitations and future work" appendix can be significantly improved. Currently, it focuses on future work and omits some limitations of the experiments, such as:
- using GPT-4 as the primary metric will make the results irreproducible once OpenAI cycles out GPT4, a closed-source model
- evaluating only on relatively weaker models (pre-previous gen, 7B) may miss some caveats or synergies from more capable LLMs
- using a single training dataset makes it possible that the proposed method is uniquely powerful in this one scenario but not others

The quality of the limitation section did not affect my score.

---

> ### Author Rebuttal · Authors · 2024-08-07
>
> **Q1a: Choice of baselines**
>
> A: **We have compared our work with ARGS [26]**. Both [26] and [39] are controlled decoding methods. Specifically, [26] directly uses a pre-trained reward model, while [39] further trains a value function that can predict the reward from partial responses. In our paper, we refer to ARGS [26] as controlled decoding, and [39] as controlled decoding with prefix. Lines 242-247 provide a detailed description of this naming strategy. If this causes confusion, we are open to using the original names in our revised version.
>
> DeAL [22] does not provide source code and has not been published yet, making it difficult to reproduce their method. Value Augmented Sampling is a concurrent work [21].
>
> **Q1b: Choice of models**
>
> A: We want to clarify that Vicuna and Falcon are based on Llama 2 instead of LLaMA 1. Using Llama 2 family as the base model was still the most common choice for academia research when we submitted this paper since Llama3 was released just one month before the NeurIPS deadline. That being said, we are running additional experiments using Llama 3 and we expect we can share the results with you around this weekend.
>
> As for the model size, we agree that it would be ideal to test on larger models beyond 7B. However, as a small research lab in academia, we lack the computing resources to test on 70B models. Testing on a 7B model is the best we can manage with our current resources. We believe that using a 7B model for testing is also common in LLM papers from academic groups, such as in ARGS and DeAL. For example, DeAL uses Falcon 7B and MBT 7B. ARGS uses LLaMA1 7B. We will include this point in our limitations section.
>
> **Q2: Inference Speed**
>
> A: We want to clarify that RE-Control is actually much faster than controlled coding [26,39].  This is because controlled decoding needs to evaluate multiple candidate tokens and repeatedly perform forward passes through the entire reward model. In contrast, RE-Control only requires optimization through a value function, which is merely a two- or three-layer neural network, making it significantly faster.
>
> We provide the inference time under different batch sizes in the uploaded PDF in the general response. As shown, increasing the batch size reduces the discrepancy between RE-Control and the base model, becoming negligible at a batch size of 32. Furthermore, ARGS [26] (referred to as controlled decoding in the original paper) does not support batch generation, resulting in significantly slower inference speeds compared to RE-Control. Additionally, we provide the compute-performance tradeoff analysis. Please refer to the general response for a detailed discussion.
>
>
> **Q3. Main experiments are limited to one dataset and relatively small past generation LLMs, ranked by GPT-4.**
>
> We have added experimental results on the SHP dataset using Vicuna. The results show that RE-Control also outperforms the baselines on this new dataset. Please see the general response for a detailed discussion.
>
> Regarding the model choice, please refer to our answer in Q1b. Additionally, we are conducting experiments using Llama3 8B and expect to share the results this weekend.
>
> Regarding the evaluation metrics, we follow those used in ARGS, which is the most important baseline for our paper. As investigated in [69], using GPT-4 as a proxy aligns with human evaluations over 80% of the time for quality assessments, providing a scalable method to approximate human preferences. Additionally, in Section 6.4 of the DPO paper, they demonstrated that "Humans agree with GPT-4 about as much as they agree with each other" for alignment tasks. We do not use any information from GPT-4 during training, so we do not anticipate any overfitting issues related to GPT-4. In addition to the GPT-4 evaluation, we also assess average reward, coherence, and diversity.
>
> We have also conducted human evaluations. We sampled 100 response pairs from RE-Control and ARGS on HH-RLHF using Vicuna as the backbone, and then asked humans to evaluate which response was more helpful, less harmful, and of overall higher quality. An option was provided for cases where the two responses were equally good. Two volunteers participated, each evaluating 50 comparisons without knowing which model generated the responses. The results were as follows: RE-Control vs. ARGS: Win: 31%, Tie: 52%, Lose: 17%.
>
> **Q4: Definition of state.**
>
> A: We want to clarify that policy iteration requires the input to the value function to be the full state, but it does not require the control signals to be added to the full state. This means we can train the value function on the full state and backpropagate through it with respect to partial inputs at test time. We will make this clear in the final version of our paper.
>
> **Q5: Reproducibility**
>
> A: We did include the requirements.txt file in our submitted code. We will provide a more detailed README in the revised version. Additionally, we will add the link to a version of LLama2 7B that contains its original license.

---

> > ### Author Response · Authors · 2024-08-12
> > **Results on Llama3**
> >
> > Dear Reviewer XYhX,
> >
> > We have added new results using Llama3 on the SHP dataset. For more details, please refer to the general response. If you have any further questions or concerns, please feel free to let us know.

---

> > ### Comment · Reviewer_XYhX · 2024-08-14
> > **Review update**
> >
> > I apologize for a delayed response and thank authors for a detailed response. Authors have answered my questions in full, suggested reasonable updates to the paper and provided additional experiments. Based on these updates, I have increased my score in the original review.

---

> ### Author Response · Authors · 2024-08-13
> **Thanks for your review**
>
> Dear Reviewer XYhX
>
> Thank you for your time and effort in helping us improve our work. As we approach the end of the discussion period, we wanted to check in to see if you have any further questions or comments. We are more than happy to address any additional concerns you may have.
>
> Best,
>
> RE-Control authors

---

### Official Review · Reviewer_Xnva · 2024-07-12

**Soundness:** 3
**Presentation:** 4
**Contribution:** 3
**Rating:** 8
**Confidence:** 3

**Summary:**

The paper "Aligning Large Language Models with Representation Editing: A Control Perspective" proposes a method for aligning large language models (LLMs) with human objectives through representation editing. Unlike fine-tuning, which is resource-intensive and unstable, or test-time alignment techniques like prompting that rely on the original model's capabilities, this method introduces external control signals into the hidden states of a pre-trained LLM. The method treats the LLM as a discrete-time stochastic dynamical system and applies control theory to train a value function on the hidden states, optimizing control signals at test time. The experiments show that this method, named RE-CONTROL, outperforms existing test-time alignment techniques and requires fewer resources compared to fine-tuning methods.

**Strengths:**

Innovative Approach: The use of control theory to introduce control signals into the hidden states of LLMs is novel and provides a new perspective on alignment.
Resource Efficiency: RE-CONTROL is less resource-intensive than traditional fine-tuning methods, making it more practical for large-scale applications.
Empirical Success: The experiments demonstrate that RE-CONTROL outperforms existing test-time alignment methods, showing strong generalization and alignment capabilities.
Flexibility: The method offers more flexibility than prompting or guided decoding as it perturbs the representation space dynamically during the generation process

**Weaknesses:**

Complexity: The method involves sophisticated control theory and optimization techniques, which might be challenging to implement and understand for practitioners without a strong background in these areas.
Dependency on Value Function: The success of the method heavily relies on the accuracy and training of the value function, which might introduce additional challenges in terms of training and performance.

**Questions:**

What are the specific challenges encountered during the training of the value function, and how can they be mitigated?

**Limitations:**

Limited Scope: The paper primarily focuses on aligning LLMs for helpfulness and minimizing harmfulness. It might not address other important alignment objectives comprehensively.
Potential Overfitting: The reliance on a specific value function and control signals might lead to overfitting to the training data or specific tasks, limiting the method's generalizability.
Evaluation Metrics: The evaluation metrics, while comprehensive, might not capture all aspects of alignment, especially in diverse and dynamic real-world scenarios.

---

> ### Author Rebuttal · Authors · 2024-08-07
>
> **Q1: Complexity: The method involves sophisticated control theory and optimization techniques, which might be challenging to implement and understand for practitioners without a strong background in these areas.**
>
> A: Though our work is theoretically grounded, the implementation of our method is actually very simple. At training time, we only need to train a basic value function, typically a two or three-layer neural network. At test time, optimizing through the value function involves only about 10 lines of code in the forward pass of the language model.
>
> **Q2: Dependency on Value Function: The success of the method heavily relies on the accuracy and training of the value function, which might introduce additional challenges in terms of training and performance. What are the specific challenges encountered during the training of the value function, and how can they be mitigated?**
>
> A: We did not encounter any significant challenges when training the value function, as it is underpinned by the Bellman equation. One minor challenge we faced was that the learning rate could affect the quality of the value function. However, this can be easily addressed by selecting the model based on the loss on the validation set. In the uploaded PDF in the general response, we also show the validation loss during the training process of the value function, which is very smooth. This is also an indicator that our value function is reliable.
>
> **Q3. The paper primarily focuses on aligning LLMs for helpfulness and minimizing harmfulness. It might not address other important alignment objectives comprehensively.**
>
> A: We follow the experimental setup of previous work on test-time alignment (ARGS [26]), focusing on helpfulness and harmfulness, which are the most common tasks in LLM alignment literature. The main goal is to demonstrate that dynamic representation editing can outperform guided decoding methods. While we agree that testing on other alignment objectives would be interesting, we leave this for future research.
>
> **Q4. Potential Overfitting: The reliance on a specific value function and control signals might lead to overfitting to the training data or specific tasks, limiting the method's generalizability.**
>
> A: In Section 6.2, we test RE-Control and the most competitive baselines on out-of-distribution data. Figure 4 demonstrates that our method can generalize well to new input distributions beyond the training distribution of the value function.
>
> **Q5. Evaluation Metrics: The evaluation metrics, while comprehensive, might not capture all aspects of alignment, especially in diverse and dynamic real-world scenarios.**
>
> A: We follow the evaluation metrics used in previous work on guided decoding (ARGS [26]), which is the most important baseline for our method. While we agree that proposing evaluation metrics for dynamic real-world scenarios is important, it is beyond the scope of this paper.

---

> > ### Comment · Reviewer_Xnva · 2024-08-13
> >
> > Thank you so much for the detailed response, the clarification helps.

---

### Author Rebuttal · Authors · 2024-08-07

We thank the reviewers for their valuable feedback and the time they spent on our manuscript. We would like to highlight that all reviewers agree that RE-control is an innovative approach and that viewing LLM as a dynamical system is novel. Additionally, all reviewers have noted that RE-control is technically sound. We provide additional experimental results (please see the uploaded PDF) in this general response and offer individual responses to each reviewer in detail below.

**More Results on an Additional Dataset.** We conducted further experiments on the SHP dataset (https://huggingface.co/datasets/stanfordnlp/SHP) using Vicuna as the backbone. As shown in Table 1 of the uploaded PDF, RE-Control continues to outperform all the baselines on this dataset. Additionally, we are also running experiments using the latest Llama3 8B. We expect to have the results by this weekend and will keep updating them.

**Inference Speed under Different Batch Sizes:** We provide additional analysis on inference time in Figure 1, which compares the inference time under different batch sizes. As shown, increasing the batch size reduces the discrepancy between RE-Control and the base model, becoming negligible at a batch size of 32. Furthermore, ARGS [26] (referred to as controlled decoding in the original paper) does not support batch generation, resulting in significantly slower inference speeds compared to RE-Control. This is because controlled decoding needs to evaluate multiple candidate tokens and repeatedly perform forward passes through the entire reward model. In contrast, RE-Control only requires optimization through a value function, which is a simple two- or three-layer neural network, making it significantly faster.


**Compute-performance Tradeoff at Test Time:** Figure 2 illustrates the compute-performance tradeoff between RE-Control and ARGS. For RE-Control, we vary the number of iterations when optimizing through the value function at test time, while for ARGS, we adjust the number of candidate tokens. As shown, the performance of RE-Control initially increases with more computing time but eventually decreases. This decline occurs because a large number of iterations at test time can lead to reward hacking, reducing the quality of the generated sentences. As discussed in Section 6.3, this hyperparameter can be selected based on the validation set. Since ARGS does not support batch generation, its inference speed is significantly slower than RE-Control. Even when RE-Control does not use batch generation, it outperforms ARGS when using the same computing resources. For example, when the inference time is around 155 minutes, the win rate of RE-Control is 75%, while ARGS is only 62%.


**Validation Loss Curve During Training of the Value Function:** We provide the validation loss versus the number of training steps for the value function in Figure 3. As shown, the training of the value function is very smooth, indicating its reliability.

---

> ### Author Response · Authors · 2024-08-12
> **More Results on Llama3 7B**
>
> Dear reviewers:
>
> We would like to present additional results using the Llama3 7B model on the SHP dataset. The table below shows the results. As we can see, RE-Control continues to work well on Llama3.
>
> | Model                     | Diversity ↑ | Coherence ↑ | Average Reward ↑ | Win Rate (%) ↑ |
> |---------------------------|-------------|-------------|------------------|----------------|
> | Prompting                 | **0.878**   | **0.672**   | 0.878            | 55.3           |
> | Static RE + Prompting     | 0.871       | 0.669       | 0.892            | 57.0           |
> | CD + Prompting            | 0.874       | 0.664       | 0.903            | 61.6           |
> | CD Prefix + Prompting     | 0.869       | 0.668       | *0.907*          | *62.3*        |
> | Ours + Prompting          |  *0.876*     | *0.671*     | **0.921**        | **65.6**       |
>
> *Table. Performance comparison between RE-Control and other test-time alignment approaches on `SHP` using `Llama3 7B` as the backbone. The win rate is evaluated by `GPT-4` as the rate at which the model's response is rated better than the preferred response in the dataset. CD is an alias for ARGS.*

---

### Decision · Program_Chairs · 2024-09-25

**Decision:**

Accept (poster)

**Comment:**

This paper presents RE-CONTROL, a novel method for aligning Large Language Models (LLMs) with human objectives. It diverges from traditional fine-tuning approaches by introducing external control signals into the hidden states of a pre-trained LLM during test time. The method views the LLM as a discrete-time stochastic dynamical system and leverages control theory to train a value function on the hidden states. This allows for the optimization of control signals at test time, dynamically influencing the model's output.

The main contribution is the innovative use of control theory for LLM alignment. By treating LLMs as dynamical systems, the authors open a new avenue for influencing model behavior without modifying the underlying model parameters. This approach proves less resource-intensive than fine-tuning while offering more flexibility compared to prompting or guided decoding. Empirical results demonstrate the effectiveness and efficiency of RE-CONTROL. The paper is well-written.

Reviewers have raised concerns about complexity, runtime efficiency, dependencies on value functions, and limited scope. The rebuttal has provided extensive clarification and new results, addressing most of the questions. Therefore, I recommend to accept this paper.

For camera-ready, I strongly recommend the authors to include their new text and results, especially those about runtime efficiency analysis. Authors should also proofread and fix the typos in their formulas.